# Suicide Research with Refugee Communities: The Case for a Qualitative, Sociocultural, and Creative Approach

Caroline Lenette 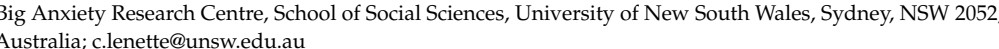

Big Anxiety Research Centre, School of Social Sciences, University of New South Wales, Sydney, NSW 2052, Australia; c.lenette@unsw.edu.au

**Abstract:** People from refugee backgrounds experience distinctively complex situations pre- and post-resettlement and are at heightened risks of suicide. The bulk of research on refugee suicide and suicidal ideation is based on diagnostic perspectives, biomedical approaches, and quantitative measures. To explore lived experience of suicide among refugee communities in more depth, this review highlights the need for qualitative, creative methods and a different paradigm to conceptualise suicide research from a social and cultural perspective as an alternative to framing and treating suicidality purely as a mental health issue. Situational and lived experience-based knowledge can significantly expand understandings of how to curb the rise in suicidal ideation and reduce suicide risks among refugees. In this context, creative research methods can be excellent tools to uncover the deeply contextual dimensions of suicidality. When interdisciplinary research explores subjective and sociocultural meanings attached to suicidal ideation, there is a greater potential to develop culturally safe supports, which are models attuned to cultural norms as determined by those most affected by lived experience of an issue or problem. Qualitative suicide research using creative methods and grounded in sociocultural knowledge can address the multidimensional and situational factors affecting refugee communities to improve interventions beyond medical framings.

**Keywords:** suicidality; sociocultural approach; critical suicide studies; qualitative research; creative methods; cultural safety; forced migration; refugees; suicide prevention; decolonial approach

## 1. Introduction

Suicide is a leading cause of death worldwide; over 700,000 individuals died by suicide in 2019 (World Health Organization 2021). Suicide and suicide attempts have profound impacts on individuals, families, and communities (Vijayakumar 2016). Each suicide is estimated to affect at least seven individuals (White et al. 2016), which means that many people around the world have lived experience of suicide. In research contexts, lived experience of suicide includes personal experiences of suicidal thoughts or attempts, caring for a suicidal person, or being bereaved or affected by suicide in any way (Queensland Transcultural Mental Health Centre 2020). Over 50% of people who die by suicide have not sought formal help (Tang et al. 2022).

Suicide risks among refugees are significant. Highly traumatic experiences pre- and post-resettlement (Colucci et al. 2017; Lenette et al. 2013) place refugee-background individuals and communities at heightened risks of suicide (Haase et al. 2022; Ingram et al. 2022). There is overwhelming evidence that forced migration experiences such as flight, loss, trauma, violence, physical abuse, torture, and feelings of hopelessness are devastating and closely linked to suicide and self-harm (Colucci et al. 2015; Vijayakumar 2016). In Australia, for instance, people from refugee backgrounds are more vulnerable to self-harm and suicidal behaviour than the general population (Dudley 2003). While suicide rates for detained asylum seekers in Australia are documented (e.g., Procter et al. 2018), data on resettled refugees are lacking because visa status is not recorded in suicide statistics (the problem of unreliable records is common in many countries; see, for instance, Canetto et al.

2023 on the case of Nepal). Where data on such trends are more readily available, they show that the incidence of suicide and suicidal ideation among refugee communities in resettlement countries such as the United States (Ao et al. 2016) and the United Kingdom (Tham et al. 2023) is on the rise.

Studies have consistently shown that minority groups have a higher risk of suicidal behaviours compared to the general population, but the bulk of suicide research worldwide continues to focus on majority populations in western countries (Forte et al. 2018). As a result, despite heightened suicide risks linked to experiences of forced migration, suicide prevention strategies that show promise in the general population might not be effective for people from refugee backgrounds because of significant contextual differences (Vijayakumar 2016). Importantly, rising global instability will likely add to the current estimates of 35.2 million refugees (United Nations High Commissioner for Refugees 2023).

While suicide research in psychology and psychiatry has yielded important knowledge in refugee studies, suicidality is a much more complex and contextual issue (Colucci et al. 2017). Suicide and suicidal ideation are multidimensional, and for many people, profoundly sociocultural events or experiences, which can be an expression of individual as well as collective suffering (Marecek and Senadheera 2023; Wexler and Gone 2016). Despite previous recognition of the importance of contextualised and nuanced framings of suicide (e.g., Durkheim's work in sociology), biomedical lenses are now dominant in framing suicide research (as well as prevention and intervention models; see Marsh 2010). Such perspectives are largely clinical-based and do not go far enough in recognising contextual factors, meaning that biomedical approaches might not be working for refugee communities. Suicide studies seem to have inherited and remained faithful to positivist concepts even though researchers continue to expand suicide studies across vastly different contexts (e.g., Falb et al. 2013; Marecek and Senadheera 2023).

Suicide research in forced migration studies requires a different paradigm as an alternative to framing and treating suicidality purely as a mental health issue (Colucci et al. 2017). Current approaches that are depoliticised and decontextualised are not helpful in this context. Qualitative, sociocultural, and creative research can open new pathways to reimagine how to study, discuss, and address one of the most challenging issues of our times. Qualitative inquiries, especially those using creative methodologies, are more likely to "prioritises cultural rather than clinical framing and trauma-sensitive facilitation rather than medicalisation" (Froggett and Bennett 2023, p. 2). This can only happen if the inquiry is completely *removed* from positivist, individualised, and biomedical framings, to instead prioritise an expansive cultural lens (Chandler et al. 2022a) that can identify cultural meanings, historical contexts, and social significance linked to lived experience of suicidality (Wexler and Gone 2016).

This thematic review highlights the need for more qualitative, sociocultural, and creative research in suicide studies that move away from clinical and pathology-focussed debates, to better recognise and address the sociocultural factors that shape lived experiences of suicide among refugee (and broader) communities. Qualitative methods are still relatively rare in suicidology (see Hjelmeland and Knizek 2010, 2016), despite the need for research models attuned to process rather than exclusively content driven (White et al. 2016). This represents a major gap in the body of knowledge on suicide, as qualitative approaches can yield deeper and broader sociocultural understandings that are currently lacking, especially in refugee research (Colucci et al. 2017). In situating this discussion in the field of critical suicidology, now referred to as critical suicide studies (see Chandler et al. 2022b), this review suggests that qualitative, sociocultural, and creative research can offer an important alternative to uncover subjective and experiential knowledge on suicide, which would be better suited and more ethical in refugee studies. As the body of knowledge on refugee suicidality is based on outsider, diagnostic or biomedical quantitative measures, research exploring lived experience and cultural knowledge using contextual approaches can expand current understandings of suicidality and better inform culturally safe suicide prevention strategies. Culturally safe models are attuned to cultural norms

and knowledge as determined by those most affected by lived experience of an issue or problem (Lenette 2019, 2022; Papps and Ramsden 1996).

## 2. Critical Suicide Studies and the Importance of Culture

The field of critical suicide studies favours a creative and compassionate approach to studying a deeply complex phenomenon that has been simplified to a medical diagnosis (White et al. 2016). Critical suicide studies question narrow biomedical understandings and instead engages with language, discourse, power relations, and social histories that frame knowledge on suicide within specific social, historical, cultural, and political contexts for a broader set of responses (see Chandler et al. 2022a, 2022b). This approach to suicide research recognises the value of contexts, lived experience expertise, and creative methods to studying a deeply complex phenomenon (White et al. 2016). Critical suicide studies open new pathways to ask *alternative* questions, which might be considered peripheral using a medical lens. Importantly, while this area of research is enriched with First Nations knowledge (e.g., Kral and Idlout 2016; Stoor et al. 2015; Wexler and Gone 2016), critical suicide studies currently lack refugee perspectives. This speaks to a gap in the literature, perhaps based on a misconception that sociocultural approaches in First Nations contexts are applicable to explorations on the lived experiences of people from refugee backgrounds.

Critical suicide studies have made significant contributions to decolonial suicide research and methodologies. A decolonial approach challenges colonial and western understandings of suicidality (X and polanco 2021). As such, a decolonial research lens is ideal to explore the social, historical, cultural, and political contexts of suicide from refugee perspectives, for more culturally responsive solutions. A key example can be found in the work of the *Big Anxiety Research Centre* (Big Anxiety Research Centre (BARC) 2023), which is a leading interdisciplinary unit in Australia where researchers aim to transform thinking and practices in mental health through creative collaboration and cultural innovation. BARC's collaboration with resettled South Sudanese young people on mental health and suicide led to the creation of a short film, *Heartsick* (feltExperience & Empathy Lab. 2022) where the group used poetry as a method to prompt discussions on intergenerational trauma and suicide (see Gitau et al. 2023—this special issue). One young woman said, "When it comes to suicide, I feel that it's a difficult thing for others to understand—why do you want to kill yourself? What's so hard in your life that you want to go through that?". Another participant shared, "It's at night that it gets to me the most and it's at night that I have those thoughts of wanting to end my life". Such creative research initiatives (here using poetry and filmmaking), point to the value of discussing suicide outside a medical framing and of co-developing culturally safe spaces for candid and meaningful conversations on the topic.

Culture, or the framework that gives meaning to societies to reproduce themselves as recognisable entities (Wexler and Gone 2016), is significant in understanding suicidal behaviours (Akotia et al. 2019; Chandler et al. 2022a; Colucci et al. 2015; Mueller et al. 2021). As Marecek and Senadheera (2023, p. 2885) argue, "[e]very culture has a canonical narrative of suicide". Even though culture is a central consideration, it remains a long-neglected topic in suicidology (White et al. 2016). Previous studies have highlighted the need for more research on how cultural values and beliefs, religion, and spirituality affect suicidality among refugee groups (e.g., Boyd and Chung 2012; Colucci et al. 2015) but there is still a gap in the knowledge on cultural meanings attached to suicide (see Chandler et al. 2022a for a discussion on the limitations and possibilities of "culture"). Recent exceptions (not related to forced migration) include Canetto et al.'s (2023) research on dominant cultural and gendered suicide attitudes and beliefs in Nepal, and Marecek and Senadheera's (2023) study of the meanings attached to suicide attempts among young girls in rural Sri Lanka in the cultural and social contexts in which they were situated. Clearly, culturally specific meanings linked to suicide differ across groups, yielding diverse cultural explanatory frameworks that can better inform approaches to addressing this issue (Stoor et al. 2015; Vijayakumar 2016). For example, cultural considerations were central to the development of Indonesia's first national suicide prevention strategy (see Onie et al. 2023).

*First Nations Models*

This review draws inspiration from models prioritising First Nations peoples' culturally grounded knowledge on suicide and social and emotional wellbeing as the aspirational ideal for a shift from deficit-based perspectives towards culturally safe suicide prevention. Such models represent a promising solution to disrupting the dominance of colonial and biomedical research lenses that merely pathologise suicidality rather than appreciate the range of contextual and sociocultural factors affecting lived experience of suicide. Significant advances can be found in First Nations models in Australia, Sweden, and Canada, where culture plays a central role in understandings of and strategies to address suicide in First Nations communities. In Australia, for instance, prevention models that prioritise the perspectives of First Nations peoples with lived experience and cultural knowledge, such as the Centre of Best Practice in Aboriginal and Torres Strait Islander Suicide Prevention framework (Dudgeon et al. 2018), are culturally safe because they focus on unique insights grounded in cultural knowledge and expertise. The perspectives of Sámi people in Sweden on cultural meanings attached to suicide are useful to develop culturally attuned suicide prevention interventions beyond geographical boundaries (i.e., applicable to Sámi communities across the Arctic region) (Stoor et al. 2015). In Canada, there are encouraging results from using participatory methodologies that value sociocultural factors in suicide research with First Nations peoples, who have the highest rates of suicide in North America (Kral and Idlout 2016). The recognition of First Nations' cultural notions and knowledge of suicide is indicative of what meaningful interventions are possible for refugee communities if suicide research and prevention models attend more carefully to the sociocultural.

## 3. Creative Methods

Creative methods are increasingly favoured in qualitative research that taps into embodied experiences and aims to engage in ethical and meaningful lived experience-based processes (Lenette 2019). In refugee studies, creative methods such as body mapping (Shanneik and Sobieczky 2023), walking interviews (O'Neill et al. 2010), or photography (Pienimäki 2021) are especially effective to explore sensitive and embodied experiences in trauma-informed ways.

However, it is still rare to find examples in suicide research that use creative methods, both in general and in the context of forced migration studies. There are only a handful of research projects that have adopted a creative approach, in addition to BARC's short film example mentioned above. One instance is Colucci and McDonough's (2019) research on the subjective benefits of using digital storytelling (an audio-visual first-person narrative where participants explore and represent life experiences that are particularly meaningful to them) to explore mental illness and suicidal behaviour with a group of migrant and refugee-background participants. Quantitative research projects are even rarer, one important exception being Liu et al.'s (2023) research using TV-based art therapy and music therapy and their effectiveness in reducing suicidal ideation among schoolgirls who survived abduction in Northern Nigeria when compared to a control group. These examples suggest that the interest in, and usefulness of, creative methods in suicide research is on the rise, with growing evidence of the potentially significant benefits and positive impacts of engaging in trauma-informed, creative endeavours rather than medicalised discussions to explore lived experience of suicide.

There are new research models that centre creative processes to explore the trauma and distress at the core of lived experience of suicide that might potentially be adapted for use in research with refugee-background participants. For instance, Froggett and Bennett (2023—this special issue) outlined how the visual matrix method, which invites associative responses to a "stimulus" (in this case, a film on lived experience of suicide, *The Invisible Edge*), can be reparative for audiences. The authors explain that "the unspeakable experience of suicidality was symbolised, re-elaborated, and released to continue its troubled life in the cultural context of the film, so that the audience could go on with theirs" (Froggett and Bennett 2023, p. 13). Applying this method to explorations of suicidality signals a need

for alternative research processes that foreground context, witnessing, support, and hope. Another example is BARC's arts-based festival in Warwick, Australia, (see Bennett et al. 2023, for details of the various creative methods used) to engage sensitively with lived experiences of trauma, distress, and (youth) suicide, following a recent and devastating suicide of a First Nations young person that affected the whole community. The BARC event fostered post-traumatic growth and connection to self, as well as a stronger sense of community.

## 4. Discussion

There is no doubt that the suicide research landscape is essentially positivist and that it has been dominated and indeed shaped by a medical, pathologising lens. But understandings of suicide need not be anchored in the medical model (Chandler et al. 2022a, 2022b). Suicide research, especially in refugee studies, requires fresh thinking and alternative approaches. As a decolonial and culturally safe approach, suicide research that values the sociocultural and uses creative methodologies in participatory ways can offer a compelling alternative. This is because critical suicide studies signal a shift away from individualised approaches to give more attention to social suffering and the histories and social conditions of people's lives (Wexler and Gone 2016; White et al. 2016).

Such models are trauma-informed and can yield insightful findings when collaborating with people from refugee backgrounds. In this context, creative methods are less intrusive and outsider-imposed than established tools such as Mollica et al.'s (1992) *Harvard Trauma Questionnaire* that is "validated" as a cross-cultural screening tool to measure trauma, torture, and post-traumatic stress disorder and translated into 35 languages. Creative methods do not rely on a "question-asker" model, usually require minimal skills (including language abilities) to participate meaningfully, and can foster a research environment where participants are more likely to exercise agency in the process (Lenette 2019; see Chandler et al. 2022b for a discussion on methodologies in critical suicide studies).

An alternative research approach with a more expansive cultural lens moves beyond individualised or medical framing to identify cultural meanings, historical contexts, and social significance linked to suicidality (Wexler and Gone 2016). The sparse examples that prioritise creative methods in suicide research mean that there is ample scope to expand this interdisciplinary area of research (see suggestions below). In fact, the need to transform suicide research and disrupt the overreliance on positivist models cannot be overstated. Creative methods represent a promising toolset that can be more effective and ethical to uncover cultural meanings, historical contexts and social significance, as Wexler and Gone (2016) state, to better appreciate and address refugee lived experience of suicide.

While this review focusses on suicide research, it is impossible to avoid pointing out the impact of research evidence on suicide prevention services. Around the world, such services are driven by imperatives to diagnose and fix individualised problems or pathology, giving priority to medical responses (Fitzpatrick 2022; White et al. 2016), which would impact how people from refugee backgrounds access (or do not access) mental health systems (Colucci et al. 2017). Given the largely positivist body of suicide research with an unquestioned health focus in refugee studies, practitioners and policymakers might not fully appreciate the sociocultural conditions that make suicide a much more complex issue for refugee communities. It is likely that there is cultural misalignment (or misfit) in interventions that favour rapid crisis responses and acute clinical interventions over models that recognise historical, cultural, community, and family trauma (Wexler and Gone 2016). This misalignment in suicide prevention is a major challenge, as Fitzpatrick (2022) explains, because of how deeply entrenched the values of the medical model are.

In contrast, when models are built on subjective and sociocultural meanings attached to suicidal ideation, such as the First Nations initiatives outlined above, there is a greater potential to develop culturally safe support to reduce suicide risks. Policy and decision-makers and practitioners committed to more effective and contextual suicide prevention, but who have had to rely on an evidence base narrowly focussed on medicalised approaches,

are constantly looking for new and innovative models and tools that are more attentive to situational factors. In Australia, for instance, national suicide interventions in past decades have made "no discernible difference" (Jorm 2019, p. 380), as suicide rates continue to rise (Australian Bureau of Statistics 2023). This is extremely concerning and requires a complete rethink of how we study, discuss, and address suicidality. What could happen if we centred a sociocultural lens to explore lived experience of suicide instead of the medical trope? The idea is not to completely replace current approaches to suicide prevention, as this would be unrealistic. Rather, the aim is to create new pathways for research that expands understandings of suicidality using different, creative methods that capture lived experience from a different angle, for multiple possibilities and strategies to emerge. Such paradigm shifts have the potential to transform research and practice, and to save lives.

## 5. Recommendations for Future Research

**More qualitative research**: As this review identifies, the dearth of research evidence based on qualitative datasets affects the quality and breadth of scholarship on suicidality, with major impacts extending to policy debates and practice. This is partly due to disciplinary gatekeepers' ongoing negative attitudes towards qualitative research (Chandler et al. 2022b; Hjelmeland and Knizek 2016). A key research priority is to continue to disrupt conceptual understandings of suicide as a purely biomedical concern (hopefully with progressive research funding models) by growing the body of knowledge on this topic from a qualitative angle. This is a call that suicidologists have echoed over the past few years (e.g., Colucci et al. 2017; Hjelmeland and Knizek 2010, 2016) to counter the overreliance on linear *explanations* (quantitative) as opposed to contextual *understandings* (qualitative) of suicide. Refugee studies are replete with quantitative and trauma-focussed data, to the detriment of qualitative and trauma-informed evidence. This means that understandings of suicide among refugee communities might be lacking important narratives that would better identify and address contextual suicide risks, and so a change in direction of suicide research in forced migration would be beneficial.

**Prioritise participatory research**: To counter the dominance of outsider-imposed gazes on lived experience of suicide among refugee communities, research with a participatory ethos, that is, which actively involves people with lived experience of the research problem from the outset and in different stages or activities (Lenette 2022) can yield very different and useful perspectives on the topic. Participatory research practices, when implemented with care and reflexivity, are ethical, trauma-informed, and culturally safe.

**Refugee perspectives in critical suicide studies**: To avoid homogenising lived experiences in critical suicide studies under the "cultural" banner (i.e., to signify anyone outside the white normative), examining distinct perspectives from those who have experienced forced migration firsthand is essential to identify more precise and contextual specificities of suicide (see for example, Chandler et al. 2022a). Such knowledge would expand and enrich critical suicide studies—and suicidology more broadly—in significant ways.

**More gender specific research**: Suicide is a deeply gendered issue (e.g., Canetto et al. 2023) as gender norms, identities, expectations, and power relations have a major bearing on suicidality (Fullagar and O'Brien 2016). However, women's suicidal ideation is considered a 'lesser issue' and largely missing from suicide debates. Gender-specific research can elucidate the nuances of sociocultural factors that impact suicidality and generate more precise scholarship to better inform interventions.

**Diverse creative methods**: As there is a multitude of creative and arts-based methods that can be used in trauma-informed and culturally safe ways in collaboration with refugee participants (Lenette 2019), including new immersive technologies such as virtual reality and artificial intelligence (Big Anxiety Research Centre (Big Anxiety Research Centre (BARC) 2023)), it is important not to speculate on or generalise about the benefits of creative approaches based on the effectiveness of one method only. Each application of each method should be context specific.

## 6. Conclusions

As suicide rates continue to rise globally, alongside unprecedented numbers of refugees that will no doubt continue to increase, the time is ripe for prioritising different approaches and methodologies in suicide studies. Alternative models that prioritise the sociocultural, and use innovative and creative practices, have the potential to significantly shift the conversation on suicide, suicidality, and suicide prevention. Such models would be better suited to exploring the lived experience of refugee individuals and communities in depth, given the complexities and multiple dimensions that characterise their lives. This thematic review makes the case for more qualitative research from a sociocultural angle to enrich scholarship on the topic, especially in relation to forced migration studies. While the field of critical suicide studies has already begun to shift debates on suicidality, there is ample scope to expand suicide research with lenses that have received little attention to date for more precise scholarship on refugee lived experience. This in turn will better inform models to address suicide risks, with major potential to save lives.

**Funding:** This research received no external funding.

**Institutional Review Board Statement:** Not applicable.

**Informed Consent Statement:** Not applicable.

**Data Availability Statement:** Not applicable.

**Acknowledgments:** As an uninvited migrant settler, I recognise the benefits that accrue to me because of the colonisation and dispossession of First Nations peoples. I pay my respects to the Traditional Custodians of the Land where this writing took place, the Bedigal People of the Eora Nation. I live, work, and play on Aboriginal Land. Sovereignty was never ceded. The violence used to take this Land continues to this day. I also thank colleagues for their advice on developing my understanding of this topic, especially Jill Bennett.

**Conflicts of Interest:** The author declares no conflict of interest.

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
