# Peer review of "Suicide Research with Refugee Communities: The Case for a Qualitative, Sociocultural, and Creative Approach"

_socsci, doi:10.3390/socsci12110637_

Round 1

Reviewer 1 Report

Comments and Suggestions for Authors

This paper offers an interesting and thoughtful thematic review of approaches to, and understanding of, suicide prevention with refugee communities. It advocates for the greater use of creative research methodologies using a sociocultural lens. The arguments in the paper are clear and logical, and draw (I am assuming) on the author's own experiences of working with refugee communities, giving the work an authoritative credibility. Any work that focus on refugees and highlights the additional vulnerabilities and challenges faced by these communities is timely, particularly in relation to suicide and suicide prevention.

My only suggestion would be to strengthen the paper by including a wider range of critical suicide studies texts. These would help underpin many of the arguments made through the paper. For example, arguments for more qualitative research into suicide were made in key papers by Heidi Hjelmeland and Birthe Knizek in the 2010s;

Hjelmeland, Heidi and Birthe Loa Knizek.  2016. “Time to Change Direction in Suicide Research.” In The International Handbook of Suicide Prevention edited by Rory C. O’Connor and Jane Pirkis, 696-709. John Wiley & Sons, Ltd. doi: 10.1002/9781118903223.ch39.

Hjelmeland, Heidi and Birthe Loa Knizek.  2011. “What Kind of Research Do We Need in Suicidology Today?” In International Handbook of Suicide Prevention: Research, Policy and Practice edited by Rory C. O’Connor, Stephen Platt and Jacki Gordon, 591-608. Oxford: John Wiley & Sons, Ltd.

Hjelmeland, Heidi and Birthe Loa Knizek. 2010. “Why We Need Qualitative Research in Suicidology.” Suicide and Life-Threatening Behavior 40 (1): 74-80. doi: 10.1521/suli.2010.40.1.74. 

In terms of sociocultural perspectives on suicide, this paper by Amy Chandler  gives something of an overview with many useful and relevant citations;

https://social-epistemology.com/2022/10/14/reimagining-suicide-research-the-limits-and-possibilities-of-suicide-cultures-amy-chandler-joe-anderson-rebecca-helman-sarah-huque-emily-yue/

In terms of what could be called the socio-politics of suicide and suicide prevention(maybe relevant to the discussion of refugee communities) then there are a few sources that might be worth following up;

Button ME, Marsh I (2019) Suicide and Social Justice: New Perspectives on the Politics of Suicide and Suicide Prevention. New York, NY: Routledge.

Fitzpatrick SJ (2016) Ethical and political implications of the turn to stories in suicide prevention. Philosophy, Psychiatry, & Psychology 23(3–4): 265–276.

Fitzpatrick SJ (2021) The moral and political economy of suicide prevention. Journal of Sociology. Epub ahead of print 1 April 2021.

China Mills' work on the 'psychopolitics' of suicide is worth reading;

Mills, C. (2018). ‘Dead people don’t claim’: A psychopolitical autopsy of UK austerity suicides. Critical Social Policy, 38(2), 302-322. https://doi.org/10.1177/0261018317726263

In terms of how suicide has come to be framed as predominantly a medical / mental health issue, dominated by positivist methodologies then see;

Marsh I (2010) Suicide: Foucault, History and Truth. Cambridge: Cambridge University Press.

This paper gives an overview of critical suicide studies that might be helpful;

https://journals.sagepub.com/doi/10.1177/13634593211061638

Amy Chandler's 'Suicide Cultures' project (https://www.ed.ac.uk/suicide-cultures) has been using creative / art-based methods in their data collection and have at least one publication in press on their methodological approach so it might be worth keeping an eye out for that.

As I said, these citations would, I think, strengthen the paper by underpinning the key arguments put forward.

Reviewer 2 Report

Comments and Suggestions for Authors

An interesting thematic review of the suicidological research and literature that is very well written. I noticed an error in the abstract at line 8. Should it not be "qualitative" rather than qualitive?

One general comment I would have is that I was expecting this review to be more directly focused on the lived experience of suicide among refugees. Instead, it was more generally focussed on the need for qualitative research from a critical suicidological perspective. This suggests, at least to me, that you may wish to tweak your title.

At lines 39 to 45, I detect an internal contradiction. The point is made that there is a lack of suicide statistics on resettled refugees. It is further noted that there is a problem of unreliable records for these statistics. Yet, it is also asserted that the incidence of suicide and suicide ideation among refugees in resettled communities in the USA and UK is on the rise. The implication here is the suicide and its ideation is a growing problem globally. If so, then, on what basis can one draw such an inference? It is purely on the fact that the number of those who are forcibly displaced in the world has been increasing steadily over the last decade?

At lines 69 to 72, it is asserted that qualitative inquries, especially those using creative methodologies, are more likely to prioritize cultural rather than clinical framing and trauma-sensitive facilitation rather than medicalisation. It is not evident why this is the case. Approaches that utilize the "social determinants of health" could also incorporate clinical and traditional medical approaches to suicide and its ideation whether it is for the general population or refugees.

It is argued that qualitative methods are rare in suicidology, line 76, and research models attuned to process are needed and that this is a major gap in the body of knowledge on suicide. Likewise, it is argued that critical suicidology is enriched with First Nations knowledge but that it lacks refugee perspectives. At line 103 it is noted that this is a gap in the literature. Nonetheless, under "Recommendations for future research" these two points do not appear to be taken up. What is called for instead are: prioritize participatory research; more gender specific research; and, diverse creative methods. Do these recommendations address these identified gaps in the literature and, if so, how?

The conclusion seem overly brief with only a passing reference to forced migration studies and no specific mention of refugees with the lived experience of suicide. The conclusions could have elaborated further in this regard and the challenges of achieving a critical suicidology that encompasses those refugees with such a lived experience. 
